# Unexpected Poor Vision within 24 h of Uneventful Phacoemulsification Surgery—A Review

**DOI:** 10.3390/jcm12010048

**Published:** 2022-12-21

**Authors:** Harry Rosen, Stephen A. Vernon

**Affiliations:** 1Charing Cross Hospital, Imperial College Healthcare NHS Trust, Fulham Palace Rd, London W6 8RF, UK; 2Ophthalmic Service, The Park Hospital, Sherwood Lodge Drive, Arnold, Nottingham NG5 8RX, UK

**Keywords:** cataract, complications of cataract surgery, paracentral acute middle maculopathy (PAMM)

## Abstract

Review on day one post uneventful phacoemulsification surgery is no longer standard practice due to the infrequency of complications when using modern cataract removal techniques. Clinicians are therefore likely to be unfamiliar with the potential causes of reduced vision when presented with a patient in the immediate postoperative period. The purpose of this review is to discuss the various differential causes of early visual loss, for the benefit of clinicians presented with similar patients in emergency care, with the use of an illustrative clinical case of paracentral acute middle maculopathy (PAMM), which recently presented to the authors. A thorough literature search on Google Scholar was conducted, and only causes of visual loss that would manifest within 24 h postoperatively were included. Complications are inherently rare in this period; however, various optical, anterior segment, lens-related and posterior segment causes have been identified and discussed. Front-line clinicians should be aware of these differentials with different mechanisms. PAMM remains to be the only cause of unexpected visual loss within this time frame that may have no abnormal findings on clinical examination.

## 1. Introduction

Patients now expect good vision immediately following modern cataract surgery, which is usually performed using phacoemulsification. Postoperative review of uneventful surgery in the first 24 h is no longer standard practice worldwide. Significant advances in the proficiency of cataract removal, not least due to modern phacoemulsification techniques, have greatly reduced the rates of serious complications requiring surgical intervention, rendering this early review less essential [1]. Clinicians have thus become increasingly unfamiliar with the differential complications that can present to the eye surgeon or to an emergency service, the day after uncomplicated cataract surgery. Herein we discuss the possible causes of unexpected visual loss within 24 h of cataract surgery, illustrated by a recent case of paracentral acute middle maculopathy (PAMM), which presented to the authors on a routine day one postoperative review.

## 2. PAMM

PAMM has been recently described as a cause of immediate poor vision following uncomplicated phacoemulsification surgery. Only a handful of cases have been described since Saraf et al. first described the condition in 2013 [2], with postoperative cases not recognised until 2017 [3]. In the postoperative context, it is characterised clinically by a dense central scotoma, presenting on the first postoperative day, often with no clinical signs on clinical examination [3,4]. The diagnosis is defined by a hyperreflective band on optical coherence tomography (OCT) confined to the retinal Inner nuclear layer (INL), with subsequent thinning and loss of the negative signal layer of the INL.

An entity existing on the same spectrum as acute macular neuroretinopathy (AMN), PAMM is believed to be caused by transient ischaemia of the deep and intermediate retinal capillary plexus, causing irreversible necrosis of the INL [2,5,6]. Only a few previous cases of PAMM have been recorded in the UK [3,7,8]. Here we present a unique case of PAMM presenting on pad removal on the first postoperative day following routine phacoemulsification in a 73 year old, where acuity improved from counting fingers to 6/9 over the first week.

## 3. Clinical Case

A 73-year-old caucasian man of slim build was referred with bilateral cataracts in June 2021. He had hypermetropic astigmatism of both eyes, 6/12 best-corrected acuity in both eyes secondary to cataracts, an axial length of 23.2 mm, and was right-eye dominant. At preoperative assessment, he had normal intraocular pressure (IOP) and normal OCT imaging (five months before surgery) of the macula and optic disc with physiological disc cups (Figure 1). He had no significant past medical history, and his only medications were intermittent tadalafil for erectile dysfunction and vitamin supplements. He had not taken tadalafil in the two weeks before surgery.

Routine phacoemulsification and lens implantation taking 15 min with no intraoperative issues took place in November 2021, the second case on a morning list with six other cases, none of whom experienced any postoperative complications. The anaesthetic used was a peribulbar anaesthetic injection of 4 mL of lignocaine 2%/bupivicaine 0.75% mixture with no additional hyalase, delivered by a consultant anaesthetist who had regularly performed such injections for >5 years. Preoperative dilation had been achieved utilising a Mydriasert insert (0.28 mg tropicamide/5.4 mg of phenylephrine hydrochloride). Although phacoemulsification powers were not routinely measured, the cataract was not dense and required short phacoemulsification times at standard settings. Infection prophylaxis was via antibiotics placed preoperatively in the irrigation bag (gentamicin 20 mg/mL—0.4 mL in 1 L and vancomycin 500 mg mixed in 10 mL water for injection—0.2 mL in 1 L), as per a protocol used for over 15 years by the surgeon, without adverse events. These were placed in the bag by the surgeon himself, together with 0.3 mL or 1 in 1000 adrenaline, all 7 bags being prepared before the operating list commenced. For postoperative IOP rise prophylaxis, apraclonidine 1% drops were used in place of the surgeon’s usual acetazolamide due to the patient’s previous adverse reaction to cotrimoxazole.

There were no immediate anaesthetic complications, chemosis, bruising, or postoperative pain, and the patient felt well throughout the postoperative period. Blood pressure measured postoperatively was within the normal range and was not significantly different from preoperatively.

The patient was reviewed on the first postoperative day as part of the surgeon’s routine practice. Upon removal of the pad, the acuity measured finger counting only, with a central scotoma, but fully preserved peripheral vision. Clinical examination including IOP and dilated fundoscopy was entirely normal, with no relative afferent pupillary defect (RAPD) or bruising. An OCT scan (Figure 2), performed with a different scanner from the preoperative scan, was difficult to perform due to poor fixation, but demonstrated a clear increased signal in the INL compared to the scan (Figure 1) performed five months previously. At one week, acuity had gradually recovered to 6/9 unaided but required considerable searching for the chart, the patient describing “a central small island of vision in a sea of darkness”.

Ten weeks later, the acuity remained at 6/9 unaided, with some subjective improvement. However, the patient continued to report a central island of vision with surrounding loss. Formal visual field assessments were variable, likely due to a difficulty in fixation (Figure 3). The total deviation plots indicate normal retinal sensitivity outside the central 10 degrees on the 24/2 field but generalised reduced sensitivity within 10 degrees on the 10/2 fields with absolute loss maximal just above fixation. OCT at this stage with the same machine as the preoperative scans (Figure 4) demonstrated a persistent hyperreflective band at the macula, replacing the negatively reflective layer of the INL, with a doughnut-shaped thinning of the intermediate retinal layers, strongly supporting the diagnosis of PAMM. The central foveal thickness had reduced from 254 microns preoperatively to 238 microns.

Subsequent carotid and vertebral artery duplex scans were unremarkable, showing minor symmetrical flow abnormalities in the internal and external carotid arteries at the bifurcation of the common carotid and normal vertebral arteries.

## 4. PAMM: Discussion

PAMM is hypothesised to be caused by a transient ischaemia of the retinal capillary plexus; however, the cause of this ischaemia remains a point of theoretical discussion. Causative factors relevant to this case include the increasingly described association with the use of local anaesthetic injections, such as with peribulbar techniques, as used here [3]. It is of note that PAMM following phacoemulsification surgery has not been described following topical or general anaesthetic. Possible explanations for the association with techniques involving the retro-ocular infusion of local anaesthetic include the loculation of anaesthetic causing local compression or a transient ocular compartment syndrome or an idiosyncratic vasoconstrictive response to the agents injected or a combination of both [3]. Anaesthetic agents, including lignocaine and bupivacaine, without the use of adrenaline, have independently been linked with reduced retinal and choroidal blood flow [9], which may have contributed. A temporary rise in IOP post phacoemulsification is a well-recognised phenomenon that may also contribute to retinal hypoperfusion [10], and this cannot be eliminated in our case. However, apraclonidine 1% prophylaxis was used in this case, which has a maximum effect between one and three hours after instillation, the time at which postoperative IOP spikes are most common [11]. In addition, the patient reported no pain following surgery, a dull ache being usual in patients who have significantly raised IOP postoperatively.

It is unlikely that tadalafil, the newer and longer-acting sibling of sildenafil, was implicated, as the patient only uses it intermittently and not within two weeks of the surgery. Nonarteritic anterior ischaemic optic neuropathy has been previously associated with the use of tadalafil and other phosphodiesterase inhibitors [12], although recent meta-analysis has questioned this association [13]. Tadalafil has a longer-acting pharmacokinetic profile than sildenafil (36 h or more), but residual blood levels are likely to have been very low after such a period of abstinence. Whether or not it contributed, ischaemic events remain a contraindication to phosphodiesterase inhibitor use in the UK as described in the British National Formulary (BNF), and a risk–benefit decision should be weighed regarding future use of the drug given the recent diagnosis of PAMM.

This case also highlights some unusual features not consistent with a typical PAMM presentation. Based on evidence from the two most recent case series describing PAMM, very few patients recover any acuity (11%, *n* = 2/19) following the development of initial classical clinical and radiological findings [3,4]. Only one patient recovered vision post phacoemulsification, who had reasonable vision (20/50) 24 h postoperatively and was relatively young (55 years); these factors may be protective and positive prognostic indicators for PAMM. Our case likewise recovered from counting fingers on the first postoperative day to 6/9 acuity at one week. This represents the oldest patient by some 20 years to have recovered vision following phacoemulsification complicated by PAMM.

Most patients suffering from PAMM (53%, *n* = 10/19 in a recent case series [3,4]) have some predisposing systemic comorbidity such as vasculopathy, hypertension, diabetes, and their ocular sequelae. Our patient had no such comorbidities. This may have offered some protectivity, aiding the recovery of acuity postoperatively.

Several studies have made the association between PAMM and COVID-19 infection [7,8], in line with the known hypercoagulable state of COVID-19 infection. This patient tested negative for COVID-19 three days preoperatively, was fully vaccinated, and was not symptomatic at the time; however, the latent effect of hypercoagulability in recently resolved infection remains an area of global study [14].

This patient is considering the removal of the cataract in the contralateral eye. Given the unexpectedly unfortunate complication that has befallen the first eye, great care and planning for the second eye must be considered. To mitigate risk, topical plus or minus intracameral anaesthetic or general anaesthesia may reduce the risk of vascular ischaemia and the theoretical risk of CRA hypoperfusion caused by injected local anaesthetic. Although there was no evidence of a significant postoperative pressure spike, more intensive postoperative IOP-lowering prophylaxis with monitoring should be considered. Although unlikely, alternative perioperative antibiotic regimes that are premixed may minimise the risk of dilution errors and antibiotic toxicity, should this have played any part in the complication of the first eye.

## 5. A Review of Visual Loss 24 h following Uncomplicated Cataract Surgery

Several independent pathologies can present as a complication of cataract surgery on the first postoperative day, despite apparently uneventful surgery. PAMM represents the only clinical diagnosis with purely radiological signs causing a severe central scotoma. Here we will discuss the possible causes of unexpectedly poor vision the day following cataract surgery and how they present to the ophthalmic surgeon or other attending clinician.

## 6. Materials and Methods

A thorough literature search was conducted using the keywords “cataract”, “phacoemulsification”, “complications”, and “day*1” using Google Scholar and PubMed. Results were refined using filters for articles published within the past 10 years. This was to maintain generalizability and relevance to modern cataract surgery, a constantly evolving and improving field. This evolving nature was reflected in the literature following a preliminary pilot search. Further sources were taken from reference lists of included papers. Only studies discussing complications of cataract surgery within 24 h of surgery were included. Causes of complications following complicated/nonroutine cataract extraction were excluded.

## 7. Literature Review

Table 1 below identifies the causes of unexpected visual loss within 24 h of routine cataract surgery. Rates, when available, are given as absolute values presenting any time after surgery; incidence within the first operative day may therefore be significantly lower.

### 7.1. Optical

Several changes along the visual axis of the eye can alter its function as a refracting apparatus and cause reduced visual acuity on the first postoperative day. The most prominent of these optical changes occur at the anterior corneal surface or at the lens.

A shallow anterior chamber secondary to wound leak will cause a change in the radius of curvature of the anterior cornea, and an apparently myopic outcome. A flat chamber from wound leakage may also be associated with secondary corneal oedema and blurred vision not correctable with a pinhole or refraction. Contemporary surgical techniques with self-sealing wounds and modern phacoemulsifiers have made wound leak a rare event, but excessive phacoemulsification energy could induce wound warpage and early leakage. Shallow or flat anterior chamber without wound leak may be a result of aqueous misdirection syndrome, where the lens/iris diaphragm is pushed forwards by the posteriorly directed aqueous. With or without raised IOP, aqueous misdirection syndrome is commoner in females and is associated with eyes with short axial length, being most common in nanophthalmos [25].

Unexpected lens dislocations can occur in 0.2% to 1.7% of cases over the course of a lifetime, with only 0.1% occurring within the first 5 years [15], and represent the most common cause for lens explantation [26]. Risk factors most commonly associated include pseudoexfoliation, prior vitreoretinal surgery, and trauma. Presentation can be variable: some are asymptomatic, while others experience visual loss, optical aberrations, and dysphotopsias, which may be present on day one following surgery. Intraoperative indicators such as improper fixation within the capsular bag, tornout capsulorhexis, or asymmetric IOL fixation can usually herald a dislocation after surgery. Insertion of the incorrect lens or lenses in the incorrect position in the eye can also mimic these effects, e.g., partially or wholly in the sulcus instead of the capsular bag. Treatment of lens dislocation or decentration often depends on the patient’s symptoms, demographics, and the type of lens in situ.

Despite constant improvements in IOL calculation, refractive surprise also remains a common cause of patient dissatisfaction in the immediate postoperative setting. One recent study of over 280,000 operations showed >1D error in 7% of cases [16]. Thorough preoperative examination, checking of biometry and IOL formulas, intraoperative technique, and aberrometry all contribute to appropriate IOL calculation and refractive outcome.

Accidental errors of incorrect lens or incorrect patient would be an obvious source of refractive surprise. Such “never-event” errors, as labelled in by the NHS in the UK, were reported 178 times (among the many hundreds of thousands [27]) in the UK over a 4-year period between 2010 and 2014 [28], but this figure is likely to be an underestimate due to reporting bias. Incorrect orientation of toric lenses will likewise cause unexpected poor vision and would be evidenced by a degree of reversibility with pinhole assessment. Correction of malalignment can be performed in the first few weeks following the original surgery and recognition of the problem early in the postoperative period is an aid to successful management. Human error remains the most common causative factor, and care in planning surgery to avoid transcription errors, particularly in the axis of astigmatism, should occur.

### 7.2. Cornea

The cornea, the anteriormost structure of the eye, is not only a crucial part of the visual optical system but is the most densely innervated structure of sensory fibres in the body [29]. As such, postoperative complications affecting the cornea tend to be painful and readily identified. Complications include abrasion, keratitis, and opacification caused by oedema.

Despite being a common presentation to primary care, corneal abrasion is a very rare self-limiting complication of cataract surgery occurring in up to 0.04% [17]. Symptoms include irritation and foreign body sensation, and the clearest sign is positive uptake on fluorescein examination. Acute-onset keratitis can theoretically present within the first 24 h after cataract surgery as either a primary herpes simplex, reactivation, or occasionally as toxic keratitis, a poorly described self-limiting reaction to surgery with negative viral polymerase chain reaction (PCR) [30]. Keratitis occurs in up to 2.25% of cases as in one study, though often less frequently [1,18]. The most common presentation is with epithelial defects such as diffuse superficial punctate erosions. Management requires early investigation with PCR and treatment with antivirals and lubrication [18].

Causes of corneal oedema in the postoperative period include raised IOP, endothelial decompensation, and toxic anterior segment syndrome (TASS). Self-limiting IOP rise is a recognised consequence of cataract surgery in the early postoperative period [31], likely comprising between 0.031% and 2.57% of cases or higher in high-risk groups [1]. Significantly raised IOP can occur in patients with glaucoma despite postoperative acetazolamide in multiple doses [32]. The clinical significance of an incidentally raised IOP immediately postoperatively remains a point of contention, as is the decision to treat pre-emptively with IOP-lowering drugs. The vast majority of IOP elevations are inconsequential, occurring within a few hours postoperatively, and the efficacy of treatment remains unproven [33]. Nevertheless, corneal oedema is a possible sequela from uncontrolled sustained IOP rise, causing subsequent endothelial dysfunction. Tan et al. reported the rate of oedema to be up to 1.3% of cases [17]. Timely treatment, or even prophylaxis, should be considered to prevent deterioration [1].

Corneal endothelial decompensation was the most common postoperative complication in a large UK-based review in 2015 at 0.59% [19]. It refers to the failure of the ion pump mechanisms conducted by the corneal endothelial cells. Failure to do so causes stromal hydration and macroscopic oedema. Following cataract surgery, causes of early decompensation are multifactorial, including phacoemulsification and IOL-related mechanical damage [34,35].

TASS is a sterile postoperative inflammatory process that can cause rapid visual loss, pain, redness, and photophobia within the first postoperative day [36], occurring in up to 0.8% of surgeries [1]. Factors distinguishing TASS from differentials such as endophthalmitis, include severe AC inflammatory reaction in the absence of conjunctival involvement, onset within 48 h of surgery, lack of vitreal involvement or opacities, e.g., on ultrasound B-scan, sterile bacterial, or fungal MCS and good response to early intensive anti-inflammatory treatment. The full mechanism of pathogenicity remains unknown but includes nonmicrobial inflammatory cascades in response to foreign material or solutions during cataract surgery [37]. Severe cases can even manifest with plasmoid aqueous, in which the aqueous forms a rigid gel with suspended cells due to substantial aqueous protein. Cases of TASS have become extremely rare in the UK following various measures to reduce risk, including thorough irrigation of reusable cannulae; however, clinicians must be aware as an alternative differential to classical endophthalmitis [19].

### 7.3. Anterior Chamber

Any opaque material in the anterior chamber will reduce the clarity of vision in the postoperative period. Causes presenting within the first operative day include hyphaema, plasmoid aqueous, and endophthalmitis.

Hyphaema is an extremely rare consequence of modern cataract surgery, occurring in 0.07–0.2% of cases [1], usually performed by trainees or occasionally as a part of uveitis–glaucoma–hyphaema syndrome (UGH), a reactionary process within the anterior chamber caused by IOL malposition chafing [38]. Patients with UGH will often present later than the first postoperative day with pain, blurred vision, and photophobia out of proportion to physical signs. Hyphaema can occasionally be clearly seen in the AC, but microhyphaema can only be visible on slit lamp microscopy. Ultrasound biomicroscopy is an essential component of diagnosis, to visualise the malpositioned IOL.

“Plasmoid” or “plastic” aqueous occurs following a decompensated breakdown of the blood-aqueous barrier leading to dense leakage of fibrin, clotting factors, and other transudates in severe states of uveitis. This presents with bright flare on slit lamp microscopy owing to its protein content. Post cataract extraction, it can occur as part of TASS, retained lens matter, or uveitis. The rate of postcataract surgery uveitis alone without end-stage plasmoid AC is low, at around 0.24% of eyes [19].

Endophthalmitis is another cause of AC opacity within the early postoperative period; however, it is now a rare postoperative finding in modern cataract care, especially since the uptake of routine intracameral antibiotics [19]. It occurs in 0.03% to 0.7% of cases (0.04% in the UK [19]) with global variation [20]. It presents with visual loss, hypopyon, and flare activity present in most cases with associated pain and redness [39]. It is particularly rare for symptoms to develop as early as one day postoperatively, with most cases occurring 3–4 days postop and certainly within 2 weeks [40].

### 7.4. Lens and Implant Related

Despite extensive improvements in the proficiency of cataract surgery in the past two decades, lens-related complications can still occur despite apparent uneventful surgery and diligent preoperative planning. Lens-related causes of reduced vision include IOL dislocation, incorrect IOL or toric axis, refractive surprise (all discussed previously), IOL opacification, or native tissue-related defects such as incomplete extraction.

Although IOL opacification most commonly occurs in the late postoperative setting, acute postoperative and even intraoperative opacification have been documented in a handful of cases globally [41]. Some were associated with pesticide use in storage facilities causing rapid lens hydration, others with calcium phosphate deposition from ophthalmic viscosurgical devices and irrigation solution instillation. Different lens materials themselves have inherent associated tendencies towards incidental opacification; however, these invariably occur after the early postoperative period. Hydrophobic acrylic lenses acquire water-based “glistenings”, which have no apparent effect on visual acuity. They are, however, prone to calcific deposits, which do require exchange or explantation due to significant visual compromise. The mechanism remains unknown but is often associated with a diabetic history or subsequent gas contact, e.g., during vitrectomy [15]. Silicone lenses are relatively inert but can suffer from silicone droplet deposition following silicone vitrectomy or calcification in patients with asteroid hyalosis. PPMA lenses are not known to suffer from any tendency toward opacification.

Posterior capsule opacification or “secondary cataract” is an extremely common complication of phacoemulsification, presenting years following surgery, owing to fibrous migration of capsule epithelial cells to the posterior pole [42]. Although recalcitrant calcified lens material centrally positioned should be observable during surgery, retention of clear posterior cortical fibres in the visual axis that hydrate in the early postoperative period could theoretically cause an early visual loss if unnoticed intraoperatively.

### 7.5. Vitreous Opacification

Complications of routine cataract surgery presenting within one day, affecting vitreous translucency, include opacification as a component of TASS or haemorrhage. Vitreous haemorrhage is now an extremally rare complication of cataract surgery since the advent of modern microincisional, phacoemulsification-based technologies [43,44].

### 7.6. Retina

Retinal pathology is a less common consequence of cataract surgery presenting in the early postoperative period. Unlike the anterior segment, posterior segment pathology is not visible without careful fundoscopy; therefore, most pathology presents due to reduced acuity as in the illustrative case. Causes of reduced vision include PAMM, macular hole, retinal detachment (RD), or undiagnosed pre-existing pathology. As demonstrated in the illustrative case above, PAMM is the only secondary pathology that presents with no physical signs or symptoms other than visual loss, requiring OCT for diagnosis.

A relatively normal retinal appearance can occur long after ischaemic diabetic maculopathy with a thinned retina on OCT scanning. This should be considered in diabetic patients in whom the full past ocular history is unknown or unclear, particularly if they have or have had poorly controlled diabetes. Likewise, patients with old retinal vein occlusion or epiretinal membrane may only be visible on imaging. Preoperative OCT scanning in dubious cases should be encouraged where the lens density permits to identify the potential causes of comorbidity, which may not be visible preoperatively. An OCT on day one can be helpful when clinical examination is inconclusive or not available preoperatively.

Macular holes are a rare postoperative occurrence; however, several cases have been reported previously, and thus they should feature on a list of differentials for postoperative painless loss of vision. The mechanism of operative injury is likely related to tractional forces from the posterior hyaloid canal [45]. Symptoms such as metamorphopsia, positive Watzke–Allen sign, and visual acuity loss occur in the days after surgery [46]. No clear intraoperative, postoperative, or implanted-related risk factors have yet been identified.

Retinal detachment post cataract surgery can be due to iatrogenic or worsening of pre-existing detachments. The severalfold increase in risk of RD following cataract surgery refers to late-onset RD in the months to years following surgery; however, any intraocular surgery, especially when complicated by posterior capsule rupture or anaesthetic needle injury, carries a risk of acute retinal detachment. Retrospective data from multiple large-scale studies indicate that the rate is slowly decreasing, the most recent being one study in Singapore, putting the rate of RD at 0.16% of surgeries [21]. Factors such as posterior capsule rupture, male sex, younger age, myopia, and posterior vitreous detachment have all been associated with a higher risk. These specific groups would perhaps warrant adequate safety netting and closer observation postoperatively [47]. A classical presentation of RD would be of sudden painless visual loss with flashing or floaters and progressive field loss, and physical signs of RD would be visible on thorough examination of the fundus, including Schaffer’s sign for vitreous pigment or haemorrhage.

Anaesthetic-needle-related complications provide another potential source for early postoperative visual loss, if not noticed in the perioperative period. Retro or peribulbar anaesthetic carries a greater risk than subtenons anaesthetic for complications, especially penetrating globe injury and peribulbar haemorrhage, particularly if carried out by inadequately trained clinicians. Thankfully, such injuries are exceedingly rare at around 0.006% for subtenons and 0.03–0.045% for peribulbar and retrobulbar techniques, respectively [22]. Secondary pathologies from penetrating trauma include RD, intraretinal, or vitreous haemorrhage.

Suprachoroidal haemorrhage is a well-known complication of intraocular surgery caused by hypotonic rupture of the long or short ciliary arteries between the sclera and the choroid. It most commonly occurs in the intraoperative or immediate postoperative setting (in the case of cataract surgery) [23] presenting with signs of posterior segment haemorrhage such as reduced acuity, severe pain due to compartment compression, anterior chamber shallowing, loss of red reflex, raised IOP, or even vitreous prolapse into the AC. The most recent incidence rates have been reported between 0.03% and 0.13%; however, these remain at least 20 years out of date. Risk factors are largely patient related, with high myopia, glaucoma, diabetes, atheroma, and hypertension all significantly increasing risk [23].

### 7.7. Optic Nerve

Optic nerve compromise the day after cataract surgery may be due to unmasked optic atrophy or ischaemic optic neuropathy. Post-cataract non-arteritic anterior ischaemic optic neuropathy (NAION) occurs in 0.05% of cases with an unknown link of causality [24]. It can present in the days after surgery, with painless scotoma (often inferior) and pale optic disc swelling with or without haemorrhage, which may be sectoral and RAPD. Interestingly, one study showed that NAION following cataract surgery strongly predicts a NAION occurring in the fellow eye (53%) following cataract surgery [48]. This should be noted for any clinician encountering sudden visual loss in a patient with previous postcataract NAION in the fellow eye. Posterior ischaemic optic neuropathy (PION) is a recognised complication of all surgical procedures, often presents bilaterally, and is more common in non-ocular surgery such as cardiac or orthopaedic surgery [49]. Distinguishable from the clinical case presented herein, PION would present with a RAPD along with painless visual loss, unlike PAMM.

### 7.8. Bilateral Visual Loss

Bilateral visual loss following cataract surgery may occur in the setting of bilateral surgery, or in the case of unilateral surgery, in patients with severe bilateral cataracts, masking bilateral disease. These would include bilateral central (cortical) pathology, cavernous sinus, or pituitary causes.

## 8. Conclusions

Clinicians are becoming increasingly unfamiliar with the causes of visual loss one day following cataract surgery; here we have reviewed the differential causes and their presentations, as an aid to clinicians reviewing such patients in emergency care. PAMM is a unique condition in its paucity of positive clinical examination findings, and should be considered as a diagnosis of exclusion, with the appropriate radiological evidence.

## Figures and Tables

**Figure 1 jcm-12-00048-f001:**
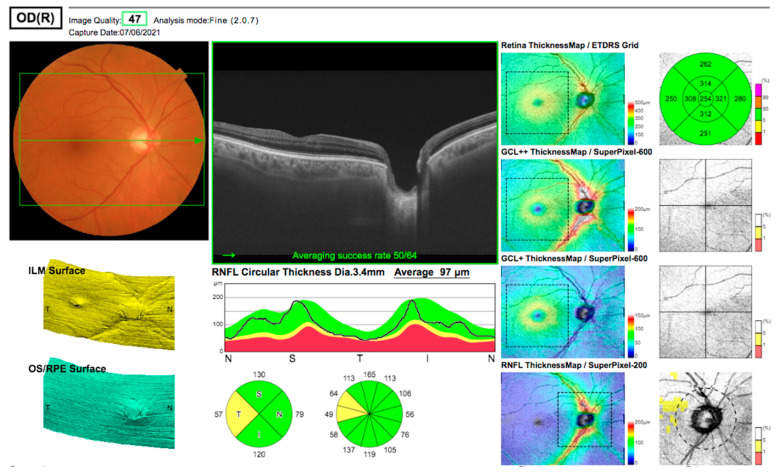
Preoperative OCT images taken on 7 June 2021. There is normal enhancement of the layers of the neural retina. Macular thickness is within the normal range.

**Figure 2 jcm-12-00048-f002:**
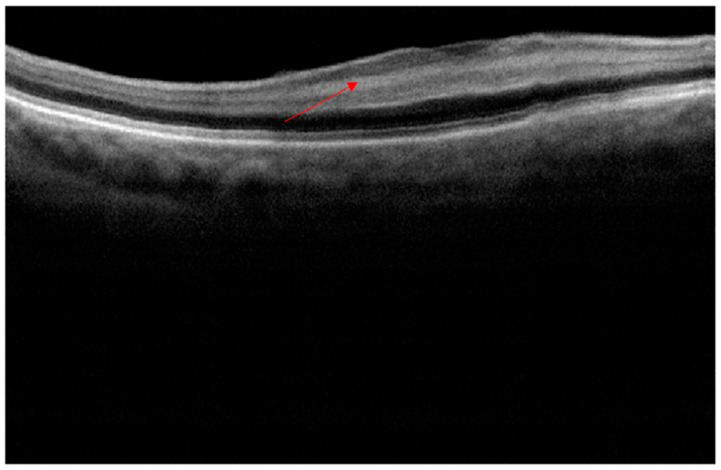
Day one postoperative OCT cross section showing total loss of the negatively enhancing inner nuclear layer, replaced by a highly reflective band marked by a red arrow.

**Figure 3 jcm-12-00048-f003:**
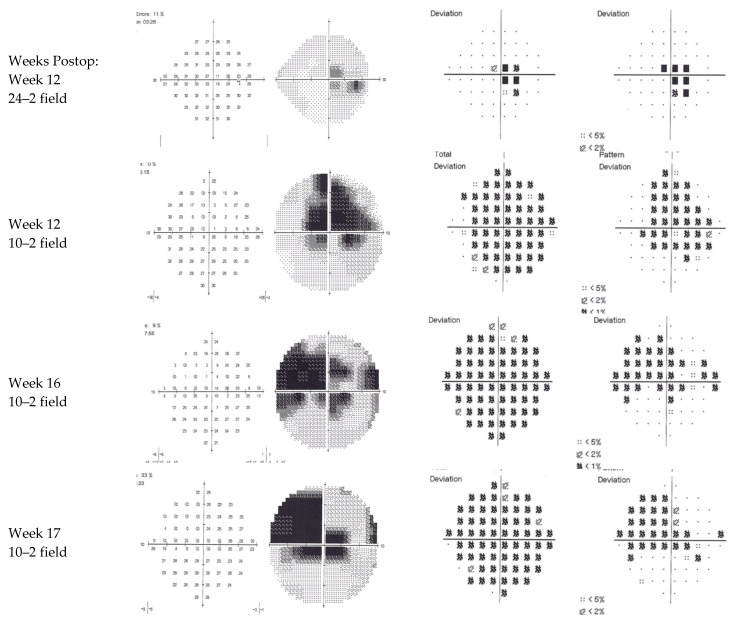
Sequential visual field assessment. The results were variable; however, they demonstrate a persistent but varying in shape scotoma on 10–2 central field perimetry.

**Figure 4 jcm-12-00048-f004:**
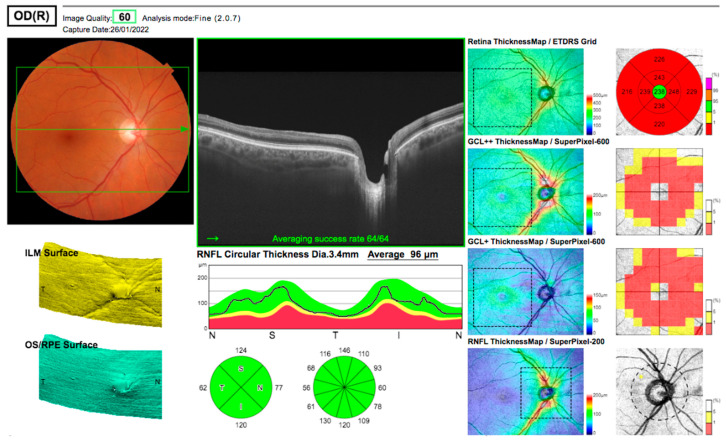
OCT imaging (26 January 2022) demonstrating persistent loss of the inner nuclear layer, with marked macular thinning but preserved circumpapillary retinal nerve fibre layer thickness.

**Table 1 jcm-12-00048-t001:** Incidence of complications causing visual loss within 24 h of uncomplicated cataract surgery.

**Optical:**	
Shallow AC from wound leak or aqueous misdirection	*
IOL dislocation	0.1–0.9% [1,15]
Incorrect IOL or axis	*
Refractive surprise >1D	Up to 7% [16]
**Cornea:**	
Abrasion	0.04% [17]
Herpes simplex keratitis	0–2.25% * [1,18]
Toxic keratitis	*
Raised IOP (corneal oedema)	0.031–2.57% * [1]
Endothelial decompensation	0.59% [19]
TASS	0–0.8% [1]
**Anterior Chamber:**	
Wound leakage	0.02–0.14% [1]
Aqueous misdirection syndrome	*
Hyphaema	0.07–0.2% [1]
Plasmoid aqueous	*
Endophthalmitis	0.03–0.7% [20]
**Lens and Implant:**	
IOL opacification	*
Residual lens matter—unnoticed at surgery	*
**Vitreous:**	
Vitreous haemorrhage or opacity	*
**Retina:**	
PAMM	*
Macular hole	*
Retinal detachment	0.16% [21]
Undiagnosed pre-existing pathology	*
Anaesthetic needle trauma	0.0006–0.045% [22]
Suprachoroidal haemorrhage	0.03–0.13% [1,23]
**Optic Nerve:**	
Unmasked optic atrophy	*
AION **	0.05% [24]
PION ***	*

* Incidence unavailable; Incidences represent absolute rates any time after cataract surgery; ** anterior ischaemic optic neuritis; *** posterior ischaemic optic neuritis.

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
