# Peer review of "Unexpected Poor Vision within 24 h of Uneventful Phacoemulsification Surgery—A Review"

_jcm, 2022, doi:10.3390/jcm12010048_

Round 1
Reviewer 1 Report
Dear Author,
I am glad to submit my review of the paper titled: "Unexpected poor vision within 24 hours of uneventful phacoemulsification surgery – a review."
The article is well-written, and the Authors should be commended for their work. Nonetheless, it requires further proofreading.
The discussion should be expanded and highlight further the possible mechanism involved in PAMM onset. Both vasoconstrictive and mechanical effects leading to a temporary reduction in the blood flow should be further evaluated and discussed.
Regarding the Literature search: Why did you choose only the last ten year of studies? Is there any reason? If yes, please clarify. Why did you choose only Google Scholar? Other sources such as Pubmed, Embase, and Scopus should be considered. Please add a table that includes the selected studies' references and main findings.
Author Response
Thank you for your helpful comments.
The discussion should be expanded and highlight further the possible mechanism involved in PAMM onset. Both vasoconstrictive and mechanical effects leading to a temporary reduction in the blood flow should be further evaluated and discussed.
We have added the following text following the highlighted original text in the PAMM discussion section - lines 128-131 to highlight the possible mixed mechanism of PAMM in the case presented
"or an idiosyncratic vasoconstrictive response to the agents injected or a combination of both"
We feel further detailed conjecture would be outside the scope of this review article and those particularly interested could read the cited articles.
"Why did you choose only the last ten years of studies?"
This was because of the evolving and nature of cataract surgery. The technique, efficacy and crucially complications of modern cataract removal surgery have changed markedly over the past few decades. The diversity of complications faced by a surgeon 20 years ago are very different to those faced by contemporary surgeons. This was reflected by the results of papers identified in preliminary pilot searches. Therefore a date restriction was felt to be the most appropriate method of ensuring generalisability and relevance to contemporary surgeons.
The following text has been added at line 189 to explain this - "This was to maintain generalizability and relevance to modern cataract surgery, a constantly evolving and improving field. This evolving nature was reflected in the literature following a preliminary pilot search."
Why did you choose only Google scholar? Other Sources such as Pubmed, Embase and Scopus should be considered.
We apologise that we omitted Pubmed as a reference source for the review. This section was written by the author who searched Google Scholar. The other author performed the Pubmed search. Pubmed has been added to the text in line 188.
Please add a table that includes the selected studies' references and main findings.
The current table already provides the most important causes of poor vision on day one following modern phacoemulsification surgery with incidences where available and has reference links. We feel that adding more information would just tend to repeat the text which is designed to be read as an anatomically based narrative in this style of article.
Reviewer 2 Report
I think this is a worthwhile addition to the general medical literature as a comprehensive guide to unexplained low vision on the first postoperative day after cataract surgery. The PAMM case is a nice lead in to the extensive review of the different causes. Notably, it refers to causes of unexplained low vision and not visual loss. I would change figure 3 to show how many days the post-op visual fields occurred instead of the date.
Author Response
Thank you for your supportive comments.
I would change figure 3 to show how many days the post-op visual fields occurred instead of the date.
We have chosen to use weeks rather than days but have altered the notes on the figure accordingly.